# Exploration into the Influencing Factors for the Intention of the Public to Vaccinate against Infectious Diseases Based on the Theory of Planned Behavior—Example of the COVID-19 Vaccine

**DOI:** 10.3390/vaccines11061092

**Published:** 2023-06-12

**Authors:** Zeming Li, Zihan Li, Xinying Sun

**Affiliations:** 1Department of Biostatistics, School of Public Health, Guangxi Medical University, Nanning 530021, China; zemingli9669@163.com; 2Department of Social Medicine and Health Education, School of Public Health, Peking University, Beijing 100191, China; 1610306117@pku.edu.cn

**Keywords:** COVID-19, infectious disease vaccines, vaccine receiving intention, TPB

## Abstract

Objectives: The coronavirus disease 2019 (COVID-19) was applied as an example, and the Theory of Planned Behavior (TPB) was used as a conceptual model. This study aimed to explore the impact of subjective norms (SNs), attitude towards the behavior (ATT), and perceived behavioral control (PBC) on the intention of the public for regular vaccination during COVID-19. The outcomes can provide certain recommendations for relevant policymakers in developing targeted health education intervention programs in the event of similar events. Methods: An online survey was performed between 17 April and 14 May 2021, via the online survey platform “WENJUANXING”. The multistage stratified cluster sampling was employed, and 2098 participants (1114 males; 53.10%) with a mean age of 31.22 years (SD = 8.29) completed the survey. The survey covered the factors influencing the public’s intention to receive future regular vaccinations during COVID-19 based on TPB. The effects of different variables on the public’s vaccination intention were analyzed using hierarchical stepwise regression. Results: The public’s intention to receive the COVID-19 vaccine (i.e., behavioral intention) in the future was taken as the dependent variable. Gender, age, marital status, education level, per capita monthly household income, vaccine-related knowledge, whether the COVID-19 vaccine was received, subjective norms (SNs), attitude towards the behavior (ATT), and perceived behavioral control (PBC) were used as independent variables. In this way, a hierarchical stepwise multiple regression model was developed. It can be seen from the final model that gender, age, vaccine-related knowledge, vaccination, attitude, SNs, and PBC were all influential factors in the public’s intention to get vaccinated in the future, with R^2^ = 0.399 and adjusted R^2^ = 0.397 (*p* < 0.001). Conclusion: TPB explains the intention of the public to receive future vaccinations to a large extent, and ATT and SNs are the most important influencing factors. It is suggested that vaccine intervention programs be developed to enhance public awareness and acceptance of vaccination. This can be achieved in three aspects, namely, improving the ATT of the public, SNs, and PBC. Furthermore, the influence of gender, age, vaccine knowledge, and previous vaccination behavior on vaccination intention should be taken into account.

## 1. Background

Infectious diseases are caused by pathogens that can be transmitted from humans to humans, animals to animals, and humans to animals [1]. Their hazards include having high infectivity and lethality, impacting productive human life, and causing social panic and uncertainty. Global public health practice has proved that vaccination can prevent, control, and eradicate infectious diseases most safely and effectively [1]. Taking the outbreak of the coronavirus disease 2019 (COVID-19) at the end of 2019 as an example, the global epidemic of COVID-19 has lasted more than two years. It has seriously endangered the life and health of humans and significantly damaged socioeconomic development. COVID-19 vaccines are now successfully used in the majority of countries. Viral mutation is a natural feature of living organisms. Most viruses, including the SARS-CoV2 virus, have the potential to mutate over time. Such mutations include the new SARS-CoV2 mutant strain identified in the United Kingdom in December 2020 [2], the SARS-CoV2 mutant strain “Delta” identified in India in October 2020 [3], the mutant strain “Omicron” first detected in South Africa on 9 November 2021, etc. [4]. The viruses Omicron [4], Omicron BA.5.2, and Omicron BF.7 had been prevalent in China since December 2022 [5]. However, a booster vaccination program for people who had received the COVID-19 vaccine at least six months previous was launched in many parts of China in October 2021 to cope with the mutated strains of the new SARS-CoV2, enhance the immunity of the population, and improve the protective effect of the COVID-19 vaccine. Due to the continuous development and mutation of the SARS-CoV2 virus, experts have concluded that the future COVID-19 epidemic may become “flu-like” and will co-exist with human beings for a long time, which means that the public may have to vaccinate against the SARS-CoV2 virus regularly or continuously in the future [6,7]. 

The theoretical models of health behavior are research-relevant in predicting vaccination intentions and are valuable tools for understanding the facilitators or inhibitors of health behavior. The theory of planned behavior (TPB) is one of the most influential in predicting health behavior [8]. It has been extensively utilized to study protective behaviors in infectious disease epidemics, such as self-isolation intentions in infectious disease epidemics [9] and the public’s intentions and actual receipt of the Hemagglutinin 1 Neuraminidase 1(H1N1) vaccine in the 2009 pandemic [10,11,12]. The theory suggests that behavioral intention is the most important influencing factor for the occurrence of health behavior, and that attitude toward behavior (ATT), subjective norms (SNs), and perceived behavioral control (PBC) can directly influence behavioral intentions and specific behavior [13]. Attitude toward behavior (ATT) means an individual’s positive or negative attitude towards the behavior to be adopted and the evaluation of beliefs, such as behavioral beliefs and outcomes. Subjective norms (SNs) indicate individuals’ subjective perceptions of the social pressure to adopt a behavior, such as motivation to comply and normative beliefs. Perceived behavioral control (PBC) refers to individuals’ perceptions of the ability to control behavior, i.e., the expected ease of performing a behavior, including control beliefs and perceived ability. The structure of TPB is shown in Figure 1. In TPB, ATT, SNs, and PBC are three significant variables that determine behavioral intentions, with more positive attitudes and support from important others and PBC being associated with more excellent behavioral intentions and vice versa.

Despite having been used to examine protective behaviors (e.g., mask-wearing and social distancing) [15,16] and vaccination intentions [17,18], TPB has seldom been utilized to explore the influence of the public on “future” intentions to vaccinate against the SARS-CoV2 virus regularly when the COVID-19 vaccine is already in use. Therefore, TPB was applied to analyze the influencing factors for the intention of the public to vaccinate against the SARS-CoV2 virus in this study and provide recommendations for developing targeted health education intervention programs and recommendations for relevant policymakers in the event of similar events. 

## 2. Materials and Methods

### 2.1. Participants and Procedures

An online questionnaire survey was conducted via WENJUANXING (wjx.cn, accessed on 17 April 2023) from 17 to 22 April 2021. Based on the sample service of WENJUANXING, a stratified random sampling method was employed according to region, age, and gender. Adults aged 18 or above were selected to participate in the survey. As structural equation modeling guarantees a sample size to observed variable ratio of at least 10:1, with 29 observed variables in this study, a minimum of 290 subjects would be required. Questionnaires were cleaned and collated, and 2098 valid questionnaires were finally acquired. The study protocol was reviewed by the Peking University Biomedical Ethics Committee (No. IRB00001052-20081) and gained its approval. All participants understood the project’s content and signed an informed consent form.

### 2.2. Measures

The questionnaire contained two parts. The first part contained demographic information, such as gender, age, marital status, education level, per capita monthly household income, vaccine knowledge, and whether the COVID-19 vaccine was administered. The second part was a scale designed by our research team based on TPB using a Likert 5-point scale. The scale was subjected to the Kaiser–Meyer–Olkin (KMO) test, with a result of 0.916. Seven factors were analyzed for exploratory factors, with a cumulative contribution of 60.74%. The internal consistency alpha coefficients for the TPB scale were acceptable, and are as follows:

**(I) ATT** includes behavioral beliefs and the evaluation of behavioral outcomes. Behavioral beliefs were assessed using four items (sample item: “Do you agree that vaccination against COVID-19 is effective in reducing the risk of SARS-CoV2 infection”) to assess individuals’ beliefs about specific behaviors with an internal consistency alpha coefficient of 0.611. The behavioral outcomes were evaluated using four items (sample item: “Do you think it is essential to lower your risk of contracting the SARS-CoV2 infection”) to assess others’ ratings of these beliefs, with an internal consistency alpha coefficient of 0.629.

**(II) SNs** include normative beliefs and motivation to comply. Normative beliefs used four items (sample item: “People important to me, including family, friends, leaders, and doctors think I should get the COVID-19 vaccine”) to assess the beliefs of an individual about the expectations of important others for the performance of specific behaviors, with an α value of 0.698. Motivation to comply was assessed using four items (sample item: “I would take the advice of someone important to me to receive the COVID-19 vaccine”) to assess the motivation of an individual to comply with significant others (including family, friends, superiors, and doctors), with an α value of 0.753.

**(III) PBC** includes control beliefs and perceived power. Control beliefs used eight items (sample item: “I will get the COVID-19 vaccine despite no new confirmed cases in my city”) to reflect an individual’s perception of any help or barriers in performing a given behavior, with an α value of 0.748. Perceived power used two items (sample item: “Vaccination is too much trouble and will take time and effort”) to assess the ability of an individual to address barriers, with an α value of 0.616.

**(IV) The behavioral intention** was assessed using three items (sample item: “If you needed to get the future COVID-19 vaccine regularly, would you receive it?”) to assess the propensity of an individual to act on a particular behavior, with an α value of 0.761.

The internal consistency Cronbach’s α coefficient for all dimensions of the TPB scale was above 0.6, and all reliability coefficients were within an acceptable range, indicating that the scale has good reliability.

The questionnaire set the quality control questions. Responses that were not logical or all options with the same answer were treated as invalid. Each identity account could only be answered once. Those submitting valid questionnaires were given a gift as a token of appreciation.

Statistical Product and Service Solutions (SPSS) 26.0 was used for statistical analysis in this study. (1) General sociodemographic variables were expressed as percentages, and the scores of each dimension of the scale were indicated by mean and standard deviation. A *t*-test or ANOVA was used to compare different sociodemographic characteristics between groups. (2) To solve the problem of scaling in the scale and make the measured entries homogeneous, the means of the product sum of corresponding entries in each variable of TPB were then squared [19,20] to analyze the influencing factors for the intention of the public to receive future COVID-19 vaccination through hierarchical stepwise regression. The test level was α = 0.05.

## 3. Results

In this study, the 2098 participants had an evaluation age of 31.22 ± 8.29, including 1114 males (53.10%). The ratio of males to females was 1.13:1. Regarding education level, most participants were undergraduates (82.84%). Concerning marital status, participants were mainly married (61.92%). Most participants had a per capita monthly household income of CNY 5000–9999 (31.02%). During the survey period, 59.01% of the survey participants did not vaccinate against the SARS-CoV2 virus, as shown in Table 1.

As shown in Table 2, it was found that men scored higher than women in SNs (*t* = 2.814, *p* < 0.05), PBC (*t* = 2.477, *p* < 0.05), and behavioral intention (*t* = 2.739, *p* < 0.05) by comparing participants with different demographic characteristics. The married group had higher scores than the unmarried one in ATT (*t* = −3.745, *p* < 0.001), SNs (*t* = −4.629, *p* < 0.001), and PBC (*t* = −7.103, *p* < 0.001). SNs (*F* = 2.780, *p* < 0.05) and PBC (*F* = 9.376, *p* < 0.001) showed an “n” trend. The higher the level of education was, the higher the vaccination intention of the participants would be (*F* = 5.531, *p* < 0.05). Participants who had already received the COVID-19 vaccine scored higher than the unvaccinated group in ATT (*F* = −5.037, *p* < 0.001), SNs (*F* = −10.669, *p* < 0.001), PBC (*F* = −12.453, *p* < 0.001), and behavioral intention (*F* = −9.829, *p* < 0.001).

Table 3 shows the hierarchical regression analysis used for progressively analyzing the prediction level of each variable of general sociodemographic factors, the previous COVID-19 vaccination status, and TPB components on the intention of the public to receive the future COVID-19 vaccine. The intention of the survey respondents to receive the future COVID-19 vaccine was used as the dependent variable. In addition, gender, age groups, marital status, education level, per capita monthly household income, vaccine-related knowledge, vaccination status, ATT, SNs, and PBC were used as independent variables. This means a hierarchical stepwise multiple regression model was developed at the α = 0.05 level. In the final model, seven variables, including gender, age, vaccine-related knowledge, marital status (yes or no), ATT, SNs, and PBC, were statistically significant with R^2^ = 0.399 and adjusted R^2^ = 0.397 (*p* < 0.001).

## 4. Discussion

This study was conducted in April 2021 when the mass vaccination with the COVID-19 vaccine began for the masses. Additionally, the influencing factors for the intention of the public to obtain future vaccinations were analyzed through the TPB. The TPB theorizes that people’s intention to behave in a certain way can directly influence the occurrence of the behavior, and behavioral intention is usually based on three factors: ATT, SNs, and PBC. In addition, TPB can better explain vaccination behavior and intentions at a micro level [21], with higher acceptance of vaccination behavior when individuals hold more positive attitudes towards vaccines [22,23,24], feel more pressure from society or others [25,26], and have more control [27,28].

ATT, SNs, and PBC explained 34.5% of vaccination intention in this study. Of the three considerations, subjective norms played the most important role, followed by ATT and PBC. SNs was the most critical factor that influenced the intention of the public to receive the COVID-19 vaccine (β = 0.423) in the future, which is consistent with the findings of Shmueli [18]. SNs respond to the influence of significant others or groups on an individual’s behavioral decisions. Simply, SNs refer to the social constraints and norms that a person feels in the society in which he or she lives to be able to engage in a particular behavior, which implies that pressure from others, including family, friends, supervisors, or doctors, will be the most important driver of future vaccination.

Attitude toward behavior involves the positive attitudes and evaluations an individual holds toward a behavior, and when a person holds strong beliefs about a behavior and has a justified evaluation of the outcome of that behavior, the more positively the person’s attitude toward that behavior can be predicted. Corace [29] argued that vaccination is a relatively complicated behavior and must be comprehended in terms of theoretical multi-factor components such as ATT and beliefs from the TPB. Guidry [30] found that adults who were more positive towards vaccination and exhibited higher levels of SNs were more likely to be vaccinated with the COVID-19 vaccine. In this study, more positive attitudes the public held towards vaccination strongly influenced behavioral intention. 

Perceived behavioral control shows that individuals’ perceptions of the ease of vaccination have a relatively small but still influential effect on vaccination intention and reflect the individual’s perception of factors that facilitate or hinder the implementation of the behavior. The effect of PBC on vaccination intention in this study was relatively small but still influential. Zhang and his colleagues [31] found that high-risk occupational groups with greater PBC had a higher vaccination intention for the COVID-19 vaccine in China. In summary, based on the results of this study, it is suggested that the publicity and education efforts of future COVID-19 vaccinations could focus on improving the intention of the public to vaccinate against infectious diseases by considering both SNs and ATT and thus increasing vaccination rates [32].

Socio-demographic factors in this study could explain the vaccination intention of the public to some degree, with female [33,34], older age [18], and less educated having a lower vaccination intention for the COVID-19 vaccine, which imply that when a participant was male, younger, and more educated, his or her willingness to receive the COVID-19 vaccine in the future also increased. Previous studies have shown that vaccination experience [35] and the acquisition of relevant knowledge [36,37] could influence the vaccination intention of the public. The extent to which vaccination experience and vaccine-related knowledge explained behavioral intentions in this study was 4.8%, which indicates that groups who had not received the COVID-19 vaccine and had a lower reservoir of vaccine-related knowledge were less likely to receive the future COVID-19 vaccine regularly.

Theories associated with behavioral science provide a perspective for learning about the influencing factors for people’s behavior. Based on the TPB, therefore, the influencing factors for the vaccination behavior of the public during the COVID-19 pandemic were explored through theoretical and empirical analyses, and recommendations were provided to inform future vaccination efforts and the development of related policies, as described below.

First, individuals should be encouraged to engage with their friends and family in health education about vaccinating against infectious diseases. Through advocacy from family, friends, colleagues, or superiors, sharing positive views and experiences of vaccination against infectious diseases will help the public understand the risk of pandemics and raise awareness of early prevention when faced with pandemic risks.

Second, developing public health intervention plans should attach importance to raising public awareness of the benefits of vaccination and the severity of the disease to increase public compliance with public health control measures [38].

Third, the development of public health intervention plans should also consider other cues for public action, such as investing more resources in publicity campaigns conducted by health-related departments to enhance public awareness and acceptance of vaccination. In addition, vaccination services should be provided in the workplace, and the public should be helped to overcome difficulties such as cumbersome vaccination processes and the lack of time or arrangements to enhance public confidence in vaccination. 

Last but not least, appropriate intervention programs should be established to address groups with low vaccination intentions to ensure actual vaccination coverage, particularly among groups with high risk. Specifically, more attention should be paid to female, older, and less educated groups with less vaccine knowledge in future vaccine intervention programs.

## 5. Conclusions

Global public health practice has proven that vaccination is one of the safest and most effective measures to prevent, control, and eradicate infectious diseases, especially during viral pandemics when newly developed vaccines can cause “vaccine hesitation” for various reasons. However, the reality is that COVID-19 continues to develop and mutate, and the public will likely be vaccinated regularly. However, various factors may still influence an individual’s intention to be vaccinated when facing a pandemic risk. In this study, we try to show that TPB provides a useful conceptual framework for vaccination intention when the public is facing a pandemic risk. We investigated how ATT, SNs, and PBC influenced the public’s intention to receive future COVID-19 vaccines at the beginning of the COVID-19 vaccine's development and use. The results show that ATT, SNs, and PBC significantly positively influence vaccination intention, with SNs playing the most important role. Furthermore, we found that socio-demographic factors such as gender, age, and education level have explanatory effects on public vaccination intention, with females, older age, and less educated groups having lower intention to vaccinate against COVID-19. Based on these findings, we provide several recommendations for public health policy.

## 6. Limitations

There are several limitations in the present study. First, this study used self-reports as the primary data collection method. Therefore, methodological biases are associated with this type of data collection, such as selection bias and recall bias. Notably, as we conducted an online survey, the elderly or those who do not use smartphones may have been excluded. Second, the design of this study was cross-sectional, and the causal relationship among the studied constructs (i.e., intention to vaccinate against COVID-19 and TPB constructs) could not be determined. Future studies should follow up with these respondents to investigate if they followed their behavioral intentions and how their attitudes and perceptions changed over time. Third, this study did not assess barriers associated with (future) reluctance to receive the COVID-19 vaccine. Therefore, it is unclear whether there are important factors that may prevent individuals from receiving the COVID-19 vaccine in the future. Therefore, future studies must focus on barriers that influence individuals to receive the COVID-19 vaccine.

## Figures and Tables

**Figure 1 vaccines-11-01092-f001:**
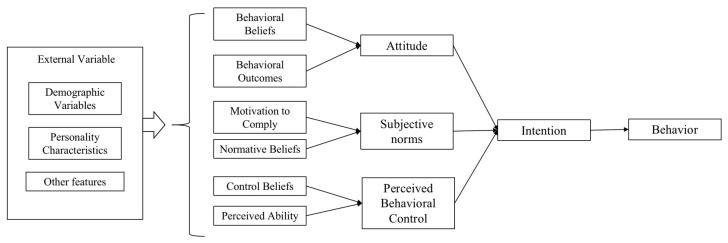
Framework diagram of TPB [14].

**Table 1 vaccines-11-01092-t001:** The demographic characteristics of the participants.

Characteristics		*n*	%
Gender	Male	1114	53.10
	Female	984	46.90
Age group	18–29	928	44.23
	30–39	862	41.09
	40–49	238	11.34
	≥50	70	3.34
Education levels	High School	180	8.58
	University	1738	82.84
	Postgraduate	180	8.58
Marital status	Unmarried	799	38.08
	Married	1299	61.92
Per capita monthly household income (yuan)	<2999	154	7.34
	3000–4999	370	17.64
	5000–9999	678	32.32
	10,000–14,999	407	19.40
	≥15,000	489	23.31
Previous COVID-19 vaccination behavior	Not vaccinated	1238	59.01
	Vaccinated	860	40.99

**Table 2 vaccines-11-01092-t002:** Scores of variables in TPB.

Characteristics	ATT	SNs	PBC	Intention
Gender				
Male	4.40 ± 0.44	4.09 ± 0.57	3.70 ± 0.73	4.52 ± 0.54
Female	4.40 ± 0.38	4.02 ± 0.59	3.62 ± 0.74	4.45 ± 0.59
*t*	0.004	2.814	2.477	2.739
*p*	0.997	0.005	0.013	0.006
Age groups				
18–29	4.38 ± 0.39	4.02 ± 0.57	3.58 ± 0.74	4.51 ± 0.52
30–39	4.40 ± 0.42	4.10 ± 0.56	3.73 ± 0.71	4.47 ± 0.56
40–49	4.43 ± 0.46	4.05 ± 0.69	3.79 ± 0.74	4.46 ± 0.68
≥50	4.45 ± 0.42	4.00 ± 0.59	3.61 ± 0.68	4.37 ± 0.72
*F*	1.524	2.780	9.376	1.728
*p*	0.206	0.041	<0.001	0.161
Education levels				
High School	4.38 ± 0.46	3.97 ± 0.68	3.59 ± 0.79	4.39 ± 0.63
University	4.40 ± 0.41	4.06 ± 0.57	3.67 ± 0.73	4.48 ± 0.57
Postgraduate	4.43 ± 0.37	4.07 ± 0.51	3.71 ± 0.71	4.58 ± 0.44
*F*	0.679	1.592	1.350	5.531
*p*	0.507	0.205	0.259	0.004
Marital status				
Unmarried	4.35 ± 0.41	3.98 ± 0.60	3.52 ± 0.75	4.48 ± 0.59
Married	4.42 ± 0.41	4.10 ± 0.56	3.75 ± 0.71	4.49 ± 0.54
*t*	−3.745	−4.629	−7.103	−0.536
*p*	<0.001	<0.001	<0.001	0.592
Per capita monthly household income (yuan)				
<2999	4.34 ± 0.38	4.01 ± 0.54	3.53 ± 0.71	4.53 ± 0.51
3000–4999	4.40 ± 0.43	4.05 ± 0.58	3.60 ± 0.76	4.46 ± 0.58
5000–9999	4.39 ± 0.39	4.07 ± 0.56	3.64 ± 0.73	4.48 ± 0.57
10,000–14,999	4.40 ± 0.40	4.05 ± 0.61	3.73 ± 0.72	4.47 ± 0.58
≥15,000	4.42 ± 0.44	4.07 ± 0.58	3.73 ± 0.72	4.51 ± 0.54
*F*	1.218	0.419	4.174	0.654
*P*	0.301	0.795	0.002	0.624
Previous COVID-19 vaccination behavior				
Not vaccinated	4.36 ± 0.43	3.95 ± 0.62	3.51 ± 0.74	4.39 ± 0.62
Vaccinated	4.45 ± 0.38	4.21 ± 0.48	3.89 ± 0.65	4.62 ± 0.44
*t*	−5.037	−10.669	−12.453	−9.829
*p*	0.000	<0.001	<0.001	<0.001

Note: ATT: attitudes towards the behavior; SNs: subjective norms; PBC: perceived behavioral control.

**Table 3 vaccines-11-01092-t003:** Hierarchical regression analysis predicting degree of intention to receive the future COVID-19 vaccine while controlling for demographic characteristics (*n* = 2098).

Variable	Modeling 1		Modeling 2		Modeling 3	
	B	β	B	β	B	β
Constant	4.441 **		4.074 **		1.300 **	
Gender	−0.074 *	−0.066 *	−0.074 *	−0.066 *	−0.044 *	−0.039 *
Educational level	0.086 *	0.063 *	0.030	0.022	0.033	0.025
Age group	−0.061 *	−0.085 *	−0.068 **	−0.095 **	−0.048 *	−0.067 *
Marital status	0.071 *	0.062 *	0.052	0.045	−0.038	−0.033
Per capita monthly household income	−0.008	−0.018	−0.011	−0.024	−0.005	−0.011
Vaccine knowledge			0.039 **	0.112 **	0.01 7 *	0.049 *
Previous COVID-19 vaccination behavior			0.204 **	0.178 **	0.059 *	0.052 *
ATT					0.265 **	0.194 **
SN					0.411 **	0.423 **
PBC					0.079 **	0.103 **
ΔR^2^	0.013		0.048		0.345	
R^2^	0.013		0.061		0.399	
adjusted R^2^	0.011		0.058		0.397	

* *p* < 0.05, ** *p* < 0.01. Indication: gender: 1 = male, 2 = female; education level: 1 = high school, 2 = university, 3 = postgraduate; age: 1 = 18–29, 2 = 30–39, 3 = 40–49, 4 = ≥50; marital status: 1 = unmarried, 2 = married; per capita monthly household income: 1 = <2999, 2 = 3000–4999, 3 = 5000–9999, 4 = 10,000–14,999, 5 = ≥15,000; previous COVID-19 vaccination behavior: 1 = not vaccinated, 2 = vaccinated.

## Data Availability

Data are not available due to ethical restrictions.

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
