# Peer review of "Exploration into the Influencing Factors for the Intention of the Public to Vaccinate against Infectious Diseases Based on the Theory of Planned Behavior—Example of the COVID-19 Vaccine"

_vaccines, 2023, doi:10.3390/vaccines11061092_

Round 1
Reviewer 1 Report
This manuscript seems to be a really important paper if it will be publsihed becasue of the content relative to the topic, which is about the coronavirus disease 2019 (COVID-19). This disease was taken as an example to explore the influencing factors for the intention of the public to get vaccinated regularly in the future based on the theory of planned behavior (TPB). A scientific basis was provided for developing targeted health education intervention programs and related policies on vaccination against infectious diseases in the future. The authors clearly evidence that the intention of the public to receive the future COVID-19 vaccine (i.e., behavioral intention) could be used as the dependent variable. In addition, besides, gender, age, marital status, education level, per capita monthly household income, vaccine-related knowledge, whether the COVID-19 vaccine was received, subjective norms (SNs), attitude towards the behavior (ATT), and perceived control over the behavior (PBC) were used as independent variables. The authors, based on the outputs emerged by conducting an online questionnaire based survey, suggested that vaccine intervention programs be developed to enhance public awareness and acceptance of vaccination and increase the vaccination intention of the public in three ways: improving the ATT of the public, SNs, and PBC. Although, it is a well written manuscript, there are some parts, which need to be improved and the relative suggestions are included in the attached file. In addition, I would suggest to use at least one graphic to present some of the results included in the tables.

Regarding the english language, there just some grammatical errors/mistakes, which can be arranged by the authors.
Author Response
Response to Reviewer 1 Comments
Point 1: This manuscript seems to be a really important paper if it will be publsihed becasue of the content relative to the topic, which is about the coronavirus disease 2019 (COVID-19). This disease was taken as an example to explore the influencing factors for the intention of the public to get vaccinated regularly in the future based on the theory of planned behavior (TPB). A scientific basis was provided for developing targeted health education intervention programs and related policies on vaccination against infectious diseases in the future. The authors clearly evidence that the intention of the public to receive the future COVID-19 vaccine (i.e., behavioral intention) could be used as the dependent variable. In addition, besides, gender, age, marital status, education level, per capita monthly household income, vaccine-related knowledge, whether the COVID-19 vaccine was received, subjective norms (SNs), attitude towards the behavior (ATT), and perceived control over the behavior (PBC) were used as independent variables. The authors, based on the outputs emerged by conducting an online questionnaire based survey, suggested that vaccine intervention programs be developed to enhance public awareness and acceptance of vaccination and increase the vaccination intention of the public in three ways: improving the ATT of the public, SNs, and PBC. 1)Although, it is a well written manuscript, there are some parts, which need to be improved and the relative suggestions are included in the attached file.2) In addition, I would suggest to use at least one graphic to present some of the results included in the tables.
Response 1: Thank you for pointing this out. We have made corrections according to the Reviewer’s comments, as follows:1)Latin vocabulary like vice versa, per capita rewrite in cursive in the manuscript. 2) We have made the appropriate additions in [4. Discussion ]. First, we rewrite "for the masses began" to "began for the masses." Second, we rewrite the sentences, including the reference number [28] and [29].
Response 2: In this study, we have used hierarchical stepwise regression analysis to explore the factors influencing the public's intention to vaccinate, and based on previous studies of this type, the results of hierarchical stepwise regression analysis are usually presented using tables to compare the data. Could you tell me what kind of graphs should be used further to present the results of the hierarchical stepwise regression analysis? Could you provide some references? Your answer would benefit our research— Thank you for your good comments.
Reviewer 2 Report
Dear authors,
Abstract
1. "A scientific basis was provided for developing targeted health education intervention programs and related policies on vaccination against infectious diseases in the future." is vague. Did you come up with the scientific basis? If the scientific basis was from other research, the word "provided" might not be suitable. Please replace with "suggested". However, the sentence would still be vague and requires another supporting sentence.
2. Under Methods, please include the instrument/method used to analyse the survey.
3. Under Results, remove "Besides".
Background
1. Line 63- neuraminidase, not neurominidase.
2. Line 63- What do you mean by vaccination intentions and behaviors in H1N1? H1N1 is the virus. Do you mean the disease or the pandemic? Revise sentence.
3. Figure- where's the figure caption? Figure was also too brief and copied from other author. Advisable to omit.
Materials and methods
1. TPB is not a new technique. Please quote the previous author(s) who developed the technique.
Results
1. Tables 1 and 3 are missing the caption. Table 2 is missing the title. What are B and "beta" in the modeling? Typo for "BEHABIORAL". Overall, please improve all table captions and info.
2. "Assignment" should be changed to "Indication"
Discussion
1. "receive the future new crown vaccine" - unsuitable choice of words.
2. Why are the specific age group was selected? Using the same methods, what is predicted for the young adolescents?
References
1. Reference no. 13 could not be found in the text.
2. Improve formatting of reference list. Please mention the retrieval date for those data from the websites.
Needs to be improved.
Author Response
Response to Reviewer 2 Comments
Point 1:
Abstract
- "A scientific basis was provided for developing targeted health education intervention programs and related policies on vaccination against infectious diseases in the future." is vague. Did you come up with the scientific basis? If the scientific basis was from other research, the word "provided" might not be suitable. Please replace with "suggested". However, the sentence would still be vague and requires another supporting sentence.
- Under Methods, please include the instrument/method used to analyse the survey.
- Under Results, remove "Besides".
Response 1:
1) We have re-written this part according to the Reviewer’s suggestion, as follows:
"Objective The coronavirus disease 2019 (COVID-19) was taken as an example and utilized the Theory of Planned Behavior (TPB) as a conceptual model and aimed to explore the impact of attitude towards the behavior (ATT), subjective norms (SN), and perceived behavioral control (PBC) on the public's intention to get vaccinated regularly during COVID-19 risk. The study's outcomes might provide recommendations for developing targeted health education intervention programs and relevant policymakers in the event of similar events. "
- We have re-written Methods according to the Reviewer’s comment, as follows:
"Methods An online survey was performed between April 17 and May 14, 2021, via the online survey platform "WENJUANXING," utilizing multistage stratified cluster sampling, 2,098 participants (1,114 males; 53.10%) with a mean age of 31.22 years (SD=8.29) completed the study. The survey explored factors influencing the public's intention to receive future regular vaccinations during COVID-19 based on TPB. The effects of different variables on the public's vaccination intention were analyzed using hierarchical stepwise regression. "
- We removed "Besides" under Results.
Point 2:
Background
- Line 63- neuraminidase, not neurominidase.
- Line 63- What do you mean by vaccination intentions and behaviors in H1N1? H1N1 is the virus. Do you mean the disease or the pandemic? Revise sentence.
- Figure- where's the figure caption? Figure was also too brief and copied from other author. Advisable to omit.
Response 2:
- We have revised "neurominidase" to neuraminidase; thank you for the reminder.
- Regarding the use of H1N1 in the manuscript, we have corrected the relevant sentence in [ Background], as follows:
"The theory of planned behavior (TPB) is the most influential in predicting health behavior [8]. It has been extensively utilized to study protective behavior in infectious disease epidemics, like self-isolation intentions in infectious disease epidemics [9] and intentions and actual receipt of the 2009 Pandemic Hemagglutinin 1 Neuraminidase 1(H1N1) vaccine [10-12]. "
- We have redrawn the conceptual framework diagram of TPB, which is quoted from the book---Science of Health Education [FUH: Science of health education[M].Beijing: People's Medical Publishing House, 2019:55.] , and added the diagram title.
Point 3:
Materials and methods
- TPB is not a new technique. Please quote the previous author(s) who developed the technique.
Response 3:
1) We are very sorry for our negligence in the description of the questionnaire design in this study; we have made corrections in [2.2. Measures], as follows:
"The second part is a scale designed by our research team based on TPB, using a Likert 5-point scale."
Point 4:
Results
- Tables 1 and 3 are missing the caption. Table 2 is missing the title. What are B and "beta" in the modeling? Typo for "BEHABIORAL". Overall, please improve all table captions and info.
- "Assignment" should be changed to "Indication"
Response 4:
- We have improved all table captions and information according to the Reviewer’s comment, including the captions of Tables 1 and 3, and changed "BEHABIORAL"to "BEHAVIORAL." The relationship between B and β is that: In multiple linear regression, beta refers to the magnitude of the effect of the independent variable on the dependent variable. In a multiple linear regression model, each independent variable corresponds to a β value, representing an increase of β units in the dependent variable when that independent variable increases by one unit. B is for Regression Coefficient, which indicates the strength and direction of the relationship between the independent and dependent variables. Therefore, when we use a regression model, we usually show the values of both B and β.
- We have changed "Assignment" to "Indication" in Table 3.
Point 5:
Discussion
- "receive the future new crown vaccine" - unsuitable choice of words.
- Why are the specific age group was selected? Using the same methods, what is predicted for the young adolescents?
Response 5:
- We have changed "receive the future new crown vaccine" to "receive the future COVID-19 vaccine" in [ Discussion ] according to the Reviewer’s comment.
2) There are two reasons for using specific age groups: to refer to other scholars' literature and to facilitate classification. For young people before the age of 30, the younger age group was found to have a higher willingness to receive the COVID-19 vaccine in the future compared to other age groups, according to the results of this study. Relevant additions have been made in [4. Discussion ] of the manuscript.
Point 6:
References
- Reference no. 13 could not be found in the text.
- Improve formatting of reference list. Please mention the retrieval date for those data from the websites.
Response 6:
- Thanks to your reminder, we have added the relevant parts of Reference 13 in the Background section.
2)We have already improved the formatting of the reference list and mentioned the retrieval date.
Special thanks to you for your good comments.

Reviewer 3 Report
The article “Exploration into the Influencing Factors for the Intention of the Public to Vaccinate against Infectious Diseases Based on the Theory of Planned Behavior--Example of the COVID-19 Vac-
Cine” has been reviewed.
The authors have applied the TPB to understand and predict attitude, behavior, social pressure and willingness to vaccinate against vaccine-preventable diseases, taking as an example SARS CoV2 immunization.
Nevertheless, there are other external factors that should be investigated and that can influence individual and community behavior.
COVID-19 pandemic created a sense of emergency and heightened awareness on the power of vaccines to prevent disease and severe outcomes. Governments, health organizations and media have played an important role in promoting and reassuring vaccine safety and effectiveness. However, vaccine hesitancy and misinformation has lead to disparities in vaccine uptake.
Here are some comments on the text
background:
Replace several “discovered” for identified when it refers to the detection of new virus variants
Line 45 and also in line 78, 80 and more : Change the COVID-19 virus for SARS CoV2 virus. The first refers to the disease and the second term to the virus itself
Line 63 : and vaccination intentions and behaviors in Hemagglutinin 1 Neurominidase 1(H1N1)… Does this refer to the Influenza AH1N1pm09 ? Please state with the virus’ correct nomenclature
Include figure caption for this and reference if taken from other publication
2. Material and Methods :
Include sample size calculation
· Include link to survey platform WENJUANXING or reference to it
· KMO and cumulative contribution fir better at the results section
· Line 102 : “ vaccination with COVID-19 is effective in reducing the risk of viral infection” –change to
vaccination against COVID-19 is effective in reducing the risk of SARS CoV2 infection”
· Why are alpha values stated at the end of each question? Please clarify
· I don’t think marital status is a relevant variable to take into consideration .
· It be most informing of the variable living with at risk individuals, such as small children immunocompromised or elderly.
· The variable Past vaccination behavior makes reference to what vaccines, childhood vaccines, influenza vaccine ? please clarify
Updated info on new variants and new SARS CoV2 vaccine formulation: WHO advisers recommend switch to monovalent XBB COVID vaccine , should be included in the discussion
Statement on the antigen composition of COVID-19 vaccines (who.int)
The article “Exploration into the Influencing Factors for the Intention of the Public to Vaccinate against Infectious Diseases Based on the Theory of Planned Behavior--Example of the COVID-19 Vac-
Cine” has been reviewed.
The authors have applied the TPB to understand and predict attitude, behavior, social pressure and willingness to vaccinate against vaccine-preventable diseases, taking as an example SARS CoV2 immunization.
Nevertheless, there are other external factors that should be investigated and that can influence individual and community behavior.
COVID-19 pandemic created a sense of emergency and heightened awareness on the power of vaccines to prevent disease and severe outcomes. Governments, health organizations and media have played an important role in promoting and reassuring vaccine safety and effectiveness. However, vaccine hesitancy and misinformation has lead to disparities in vaccine uptake.
The work needs major corrections to upgrade quality of publication
Author Response
Response to Reviewer 3 Comments
Point 1:
background:
- Replace several “discovered” for identified when it refers to the detection of new virus variants
- Line 45 and also in line 78, 80 and more : Change the COVID-19 virus for SARS CoV2 virus. The first refers to the disease and the second term to the virus itself
- Line 63 : and vaccination intentions and behaviors in Hemagglutinin 1 Neurominidase 1(H1N1)… Does this refer to the Influenza AH1N1pm09 ? Please state with the virus’ correct nomenclature
4.Include figure caption for this and reference if taken from other publication
Response 1:
- According to the Reviewer's suggestion, we have changed several “discovered” to identified in [ Background].
- Thank you very much for your suggestion. As Reviewer suggested, the COVID-19 virus refers to the disease, and the SARS-CoV2 virus refers to the virus itself. We have changed the COVID-19 virus to the SARS-CoV2 virus in this manuscript.
3) We have revised "neurominidase" to neuraminidase. Thank you for the reminder.
4) We have redrawn the conceptual framework diagram of TPB, which is quoted from the book---Science of Health Education [FU H: Science of health education[M].Beijing: People's Medical Publishing House, 2019:55.], and added the diagram title.
Point 2:
Material and Methods :
- Include sample size calculation
- Include link to survey platform WENJUANXING or reference to it
3.KMO and cumulative contribution fir better at the results section
4.Line 102 : “ vaccination with COVID-19 is effective in reducing the risk of viral infection” –change to vaccination against COVID-19 is effective in reducing the risk of SARS CoV2 infection”
5.Why are alpha values stated at the end of each question? Please clarify
- I don’t think marital status is a relevant variable to take into consideration .It be most informing of the variable living with at risk individuals, such as small children immunocompromised or elderly.
- The variable Past vaccination behavior makes reference to what vaccines, childhood vaccines, influenza vaccine ? please clarify
Response 2:
- Thank you for pointing this out. Based on the reviewer's comments, we have made the appropriate additions in [1. Participants and Procedures ]section, as follows:
"As structural equation modeling guarantees a sample size to the observed variable ratio observed variable of at least 10:1, with 29 observed variables in this study, a minimum of 290 subjects would be required. "
- We have added the link to the survey platform WENJUANXINGin [1. Participants and Procedures ] section.
- Thank you for pointing this out. As the KMO and cumulative contribution fir are related to the questionnaire factor analysis, we believe it is more appropriately placed in the Materials and Methods.
- According to the Reviewer’s suggestion, we have changed“ vaccination with COVID-19 is effective in reducing the risk of viral infection” to vaccination against COVID-19 is effective in reducing the risk of SARS CoV2 infection” in section [2. Measures].
- As Reviewer's suggestion, we have made the appropriate additions in [2. Measures] section, as follows:
"The internal consistency Cronbach's α coefficient for all dimensions of the TPB Scale was above 0.6, and all reliability coefficients were within an acceptable range, indicating that the scale has good reliability."
- This study found that married individuals had higher SN/ATT/PBC scores than unmarried individuals. It is possible that from the perspective of health for couples and children, married individuals with families are more likely to take protective measures such as wearing masks and getting vaccinated to protect their family and their health during a disease pandemic. We, therefore, reserve the inclusion of 'marital status' as one of the factors influencing the public's vaccination intentions.
7) According to the Reviewer's suggestion, We have changed "Past vaccination behavior" to Previous COVID-19 vaccination behavior.
Point 3:
Other changes: Quality of English Language
Response 3:
- We checked the English grammar of the manuscript to ensure, to the maximum extent possible, that there were no errors in the use of grammar and words.
- We made appropriate changes in the discussion section and added conclusions and limitations.
Special thanks to you for your good comments.

Round 2
Reviewer 2 Report
Dear authors,
Figure 1 was not original, must cite the original source in which you mentioned it was quoted from the book---Science of Health Education [FUH: Science of health education[M].Beijing: People's Medical Publishing House, 2019:55.].
The diagram title should be TPB not PBT.
There was also no in-text mentioning of Figure 1, so what's the significance having it in the paper?
Language for abstract should be improved.
Author Response
Response to Reviewer 2 Comments 2nd
Point 1:
Figure 1 was not original, must cite the original source in which you mentioned it was quoted from the book---Science of Health Education [FUH: Science of health education[M].Beijing: People's Medical Publishing House, 2019:55.].
Response 1:
We have added the reference to the structural diagram of TPB in Figure 1, which is [14] FU H, Science of health education[M].Beijing: People's Medical Publishing House, 2019:55. Thank you for your reminder.
Point 2:
The diagram title should be TPB not PBT.
Response 2:
We are very sorry for our negligence in the diagram title; we have corrected it.
Point 3:
There was also no in-text mentioning of Figure 1, so what's the significance having it in the paper?
Response 3:
We have made corrections according to the Reviewer’s comments in [1. Background] section, as follows: “The structure of TPB is shown in Fig. 1.” Figure 1 is intended to show the structure of TPB to give the reader a better understanding of this model.
Point 4:
Language for abstract should be improved.
Response 4:
We have checked, revised, and enhanced the Language in the abstract section of the manuscript, and thank you for your suggestions.

Reviewer 3 Report
Authors have satisfactorily made suggested corrections
Author Response
Thank you very much for your comments and suggestions.